# Delivering Positive Newborn Screening Results: Cost Analysis of Existing Practice versus Innovative, Co-Designed Strategies from the ReSPoND Study

**DOI:** 10.3390/ijns8010019

**Published:** 2022-03-14

**Authors:** Francesco Fusco, Jane Chudleigh, Pru Holder, James R. Bonham, Kevin W. Southern, Alan Simpson, Louise Moody, Ellinor K. Olander, Holly Chinnery, Stephen Morris

**Affiliations:** 1Department of Public Health and Primary Care, University of Cambridge, Cambridge CB2 0SR, UK; sm2428@medschl.cam.ac.uk; 2Centre for Maternal and Child Health Research, City, University of London, London EC1V 0HB, UK; j.chudleigh@city.ac.uk (J.C.); pru.holder@city.ac.uk (P.H.); ellinor.olander.1@city.ac.uk (E.K.O.); 3Pharmacy, Diagnostics and Genetics Sheffield Children’s NHS Foundation Trust, Sheffield S10 2TH, UK; j.bonham@nhs.net; 4Women’s and Children’s Health, University of Liverpool, Liverpool L69 3BX, UK; k.w.southern@liverpool.ac.uk; 5Florence Nightingale Faculty of Nursing, Midwifery & Palliative Care, King’s College London, London SE1 8WA, UK; alan.simpson@kcl.ac.uk; 6Centre for Arts, Memory and Communities, Coventry University, Coventry CV1 5FB, UK; aa0445@coventry.ac.uk; 7Faculty of Sports, Health and Applied Science, St Mary’s University Twickenham, Twickenham TW1 4SX, UK; holly.chinnery@stmarys.ac.uk

**Keywords:** health services research, health economics, genetics

## Abstract

Although the communication pathways of Newborn Bloodspot Screening (NBS) are a delicate task, these pathways vary across different conditions and are often not evidence-based. The ReSPoND interventions were co-designed by healthcare professionals alongside parents who had received a positive NBS result for their child. To calculate the cost of these co-designed strategies and the existing communication pathways, we interviewed 71 members of the clinical and laboratory staff of the 13 English NBS laboratories in the English National Health Service. Therefore, a scenario analysis was used to compare the cost of the existing communication pathways to the co-designed strategies delivered by (i) home-visits and (ii) telecommunications. On average, the existing communication pathway cost £447.08 per infant (range: £237.12 to £628.51) or £234,872.75 (£3635.99 to £1,932,986.23) nationally. Implementing the new interventions relying on home-visits exclusively would cost on average £521.62 (£312.84 to £646.39) per infant and £297,816.03 (£4506.37 to £2,550,284.64) nationally, or £447.19 (£235.79 to £552.03) and £231,342.40 (£3923.7 to £1,922,192.22) if implemented via teleconsultations, respectively. The new strategies delivered are not likely to require additional resources compared with current practice. Further research is needed to investigate whether this investment represents good value for money for the NHS budget.

## 1. Introduction

Newborn bloodspot screening (NBS) in England covers nine conditions: cystic fibrosis (CF); sickle cell disease (SCD); congenital hypothyroidism (CHT) and six inherited metabolic diseases (IMDs), including phenylketonuria (PKU); medium-chain acyl-CoA dehydrogenase deficiency (MCADD); maple syrup urine disease (MSUD); isovaleric acidaemia (IVA); glutaric aciduria type 1(GA1) and homocystinuria (HCU). NBS aims to identify pre-symptomatic, affected babies, thus allowing for early initiation of treatment. 

Annually, almost 10,000 parents are informed of their child’s positive NBS result at around 2–8 weeks after birth, depending on the condition. Most babies with initial positive NBS results for SCD and approximately 10% of those with a positive NBS result for CF will later be confirmed as gene carriers of the disease. However, approximately 1500 of the babies will eventually be diagnosed as being affected by one of the nine life-changing conditions that are currently screened for [1,2].

Communicating NBS results is a delicate task and should be carefully and appropriately crafted to prepare for a range of outcomes, which could vary considerably given the wide range of clinical profiles of positive cases. Poor, or inappropriate, communication approaches can have an impact on children and families in the short [3,4,5,6] and long term [7]. Studies suggest distress caused can manifest as arguments between couples and apportioning of blame [3,5], alteration of life plans and inability to conduct tasks of daily living such as going to work or socialising [3], long-term alterations in parent-child relationships and mistrust and lack of confidence affecting ongoing relationships with staff. [7] There is also evidence of parents reducing their child’s interaction with others, particularly in the case of CF [3]. For these reasons, delivery of positive NBS results requires careful consideration, planning and evidence to mitigate potentially distressing or harmful consequences for parents [3,4,5,6,7]. Although some guidance is available on the content and best mode of communication between health professionals and parents, the guidelines vary across the different conditions and are often not evidence-based [8,9]. The ReSPoND study aimed to fill this gap by co-designing interventions to improve the communication of positive NBS results [10].

A process evaluation of the co-designed interventions, when used in practice in three National Health Service (NHS) provider organisations in England served by two NBS laboratories (NBSLs) that had been involved in the co-design process, was undertaken [11]. Health care professionals who had used the co-designed interventions provided positive feedback in relation to their purpose, ease of completion and felt they were useful in terms of enhancing communication of positive newborn bloodspot screening results. Parents expressed that they felt the co-designed interventions improved consistency, pacing and tailoring of information and ensured reliable information was provided to families following communication of the positive NBS result. The cost of implementing these co-designed interventions, and the existing communication pathways, is unknown, and this limits the understanding of whether providing positive NBS results using structured interventions is economically viable. The aims of this cost analysis were to (a) calculate the mean costs associated with implementing the co-designed interventions during home-visits or via telecommunications compared to existing pathways per infant with a positive test result and (b) project these costs to the national level across the English National Health Service (NHS) per annum. 

## 2. Materials and Methods

This study was part of a larger programme of work within the ReSPoND study and was approved by the London Stanmore ethics committee (17/LO/2102) [11,12,13]. Communication strategies to parents start after the involvement of the laboratory, which forwards the results to the relevant clinical team, who then communicate the results to the family. Our study found that practice varies nationally in terms of the involvement of health care staff and approaches used [12].

Although Public Health England (PHE) attempted to standardise the communication of positive NBS results by producing a national guideline, there is high-variability in the standard communication strategies currently used in England—which makes a concise description of the standard pathways problematic [13]. More details on existing communication pathways considered in our research were published previously [12,14]. To improve communication of positive NBS results, the ReSPoND interventions were co-designed by healthcare professionals alongside parents who had received a positive NBS result for their child. 

There are thirteen NBSLs in England (study sites). Data were collected from all NBSLs regarding current communication strategies to provide baseline data for existing costs associated with communication of positive NBS results [12]. The interventions were co-designed by parents and health care staff in three National Health Service provider organisations (Trusts) in England served by two Newborn Screening Laboratories (NBSLs) that process comparable numbers of positive NBS reports annually for each of the nine conditions currently included in the NBS programme [10]. The co-designed interventions included standardised procedures, such as (i) condition-specific laboratory proformas for communication of positive NBS results from the NBS laboratories to clinicians based on those developed by the Department of Clinical Chemistry and Newborn Screening at Sheffield Children’s NHS Foundation Trust; (ii) condition-specific communication checklists to communicate positive NBS results to families; and (iii) condition-specific letters/emails template to provide further information to the families immediately after communication of the initial NBS result. Further details on the co-designed interventions have been published elsewhere [10]. However, these differed from existing practice as 10 out of 13 of the NBSLs created their own templates (rather than using those available via national guidelines), which meant that there was no national, standard approach for referring positive NBS results from the NBSL to the relevant clinician [12]. Moreover, standard, national, condition-specific communication checklists and condition-specific emails/letter templates were not available to clinicians to assist with communicating positive NBS to families.

We calculated the costs incurred as a result of the following pathways: (i) the existing communication pathways; (ii) communication incorporating the co-designed interventions during home-visits (Intervention pathway (IP)—home-visit); (iii) communication incorporating the co-designed interventions using telecommunications (IP—teleconsultation) rather than face-to-face home visits. For each pathway, the costs per infant were estimated and projected to the national level by multiplying the mean cost per infant by the number of English neonates having a positive test result for each of the conditions considered by the NBS during 2018—the most recent year with available epidemiological data [15]. We used a micro-costing approach to consider the number of contacts made by health care staff to inform the parents of the positive NBS result of their infants [16]. The schematics of the communication pathways were used to count the number of contacts and activities performed using the existing communication pathways and then how this would be affected if the new interventions were introduced during home-visits (Appendix A) or using telecommunications (Appendix A). 

Our study considered the interactions within the medical team and the contacts between parents and health care staff [10]. The mean time required for each of these activities was elicited from interviews with the health care staff. Based on this information, we calculated the mean cost per infant receiving a positive test result by applying publicly available unit costs to the number of contacts registered. We adopted an NHS perspective for this part of the analysis. Then, we applied these costs to the number of infants with an initial positive NBS result for each of the nine conditions (this included true and false positives) to project the costs of implementing the communication pathways at the national level in England in 2018 [15]. The cost analysis was developed implementing three steps: (i) identification of the relevant resources, (ii) measurement of the resources used and (iii) valuation of costs.

### 2.1. Identifying the Relevant Resources

The first step of the cost analysis relied on the schematics of the communication pathways (Appendix A) to identify key members of the health care staff in the existing communication pathway, namely nurses, psychologists and health visitors. The key health care staff members involved in delivering the NBS results are listed in Table 1. Contacts requiring a face-to-face interaction with the parents were categorised by their setting (clinic or home) and, where applicable, included travel time for the health care staff. Likewise, those activities that did not require a face-to-face interaction (emails, telephone calls) were categorised by whether they happened between parents and health care staff or within the medical team. It was assumed that the time needed for a telephone call corresponded to the time for drafting an email. The additional travel costs incurred by parents are considered separately below.

### 2.2. Measuring Resource Use

Once the resources used in the existing communication pathways were identified, the number of activities were counted by examining the schematics of each communication pathway, specifically (i) existing, (ii) co-designed interventions used during home-visits and (iii) co-designed interventions used during telecommunications. Where the schematic indicated that the activity could have been performed by different types of staff, such as by either a health visitor or a nurse, an equal distribution was assumed, namely a health visitor 50% of the time and a nurse 50% of the time. The mean time required to perform each of the actions outlined in the schematics was obtained from interview transcripts and is summarised in Table 1. 

### 2.3. Valuing Costs per Pathway and National Costs

Unit costs were obtained from the Personal Social Services Research Unit (PSSRU) 2019 and the overall cost per each contact was calculated by multiplying the relevant unit cost per minute by the time spent by each staff member on each activity—which is reported in Table 1. Based on the number and type of contacts, these costs were summed to obtain the overall cost per pathway, per infant [17]. The costs per each pathway are presented as means and 95% confidence intervals (CI), estimated by resampling the data on the number of contacts per centre via non-parametric bootstrapping (5000 replicates). These estimates were projected to the national level per annum by multiplying the mean cost per infant of each strategy by the overall number of infants who had received a positive NBS result in England in 2018 [15]. 

All monetary values produced by our analyses represent costs that would have occurred between 2019 and 2020.

### 2.4. Sensitivity Analysis

The robustness of our estimates was tested by a one-way sensitivity analysis, which explored whether varying the mean time required to perform each of the actions by ±25% would have changed our conclusions regarding the cost of the interventions. 

## 3. Results

Seven newborn screening laboratory staff and 24 health care staff members were interviewed. During these interviews, the participants provided an estimate of the time needed per each contact, of which the mean is shown in Table 1, alongside the unit cost per minute and the resulting cost per contact. 

The average total cost per infant per communication pathway, namely the existing communication pathways and the co-designed interventions, was obtained by considering the cost of each study site, which is reported in Figure 1. The Appendix A show the resource use and the cost at each study site. The costs per each communication pathway were summed per each centre, which were then used to compute the average cost per communication pathway across the study centres.

None of the cost differences between the pathways was statistically significant, except for both SCD strategies (i.e., affected and carriers). The mean cost of the communication pathways ranged from £236 (SCD carrier; IP—teleconsultation) to £646 (CF affected; IP—home-visit). Although the intervention pathways were more costly than the existing communication pathways, this difference was never larger than £106 (MCAAD; IP—home-visit). For MCAAD pathways, the difference between the IPs—the home-visit pathway and the existing pathway—was driven by an increase in the number of activities delivered by nurses, namely 89% more of home-visit and 14% more of not face-to-face external interaction. Similar to MCADD, the IP–home-visit registered the highest cost across the communication strategies for SCD carriers (cost increased by 32%), CF carriers (34%) and CHT (15%). By contrast, implementing the home-visits pathway consistently across the centres for infants affected by either CF or SCD had no substantial impact on costs (less than 5%). This result is not unexpected since the exiting pathways involved contacting the parents of infants affected by CF or SCD by visiting their house, on average 1.14 and 1.00 times, respectively. For infants affected by SCD, the cost of implementing the intervention via teleconsultations (£489; 95% CI: £464 to £513) was statistically lower than via home-visits £566 (£541 to £590). For SCD carriers, the teleconsultation scenario (£236; 95% CI: £216 to £252) was less expensive than delivering the NBS results via home-visit (£313; 95% CI: £293 to 329). Assuming that the intervention pathways will be delivered to parents via home-visits and teleconsultation evenly (i.e., 50% of the results delivered via home-visits and the remainder via teleconsultation), the average cost per infant of the intervention pathways is £484, which is £37 larger than the standard communication pathway’s average cost.

Figure 2 depicts the costs projected at the national level if each of the communication pathways were applied to the England 2018 newborn population. The averages of the costs at the national level per each communication pathway are displayed in Figure 2 alongside the number of infants per each disease. 

Although the cost difference between the strategies would be negligible for IMDs, mainly due to the low numbers of infants affected by IMDs, implementing the intervention pathways via teleconsultations in infants carrying the sickle cell trait would lead to saving approximately £9000 per year. Adopting any of the intervention pathways for newborns affected by SCD would not result in any further expenditure and could reduce the NHS costs by 13% if delivered by teleconsultations. Regarding children affected by CF, our estimates showed a similar trend observed in SCD. Given the limited number of CF carriers (120 infants) compared to SCD (8152) carriers, the size of the cost differences for CF carriers was smaller than those calculated for SCD.

### Sensitivity Analysis

Varying the length of the contact by ±25% did not lead to statistically significant differences, except for the CHT standard intervention pathway and the SCD intervention pathways, which were sensitive to the duration of the not face-to-face external interactions (Appendix A).

## 4. Discussion

There is a wide variation in communication approaches for feeding back positive newborn screening results to parents across England, which is influenced by organisational, practical and contextual factors. Although there is evidence of good practice, the lack of standardisation in the process could lead to repeated and tangible harms due to inconsistent or poor communication [3,4,5,7]. Previous research conducted both nationally and internationally suggests that further guidance is necessary to ensure a more consistent approach to meet the parents and staff needs tailored to the nuances of each condition [3,5,18,19]. However, resources are limited, and careful consideration should be given to prioritise their allocation. In this analysis, we provide cost estimates of innovative, co-designed communication strategies via home-visits or telecommunications compared to the existing communication pathways to deliver NBS results to families. On average, using the new co-designed strategies would lead to a small increment of the cost per infant, approximately £37 nationally per annum. One of the consequences of the COVID-19 pandemic is that many activities originally delivered face-to-face now are provided via information communication technologies. In our study, we tried to capture these changes in the health care organisation by assuming that the NBS results could be delivered via teleconsultations. Delivering the NBS results using information communication technologies (i.e., teleconsultations) would lead to a small decrease in the total mean cost projected at the national level (i.e., the cost calculated by multiplying the costs per infant by the overall number of infants by disease) (−£3392 per annum), which could culminate into a statistically significant saving for SCD infants when compared to delivering the results via home-visits. Introducing a home-visit for all conditions in all the innovative pathways would result in an additional cost of £55,005 per annum. The sensitivity analysis showed that our results were not affected by different assumptions on the duration of the activities needed to implement the communication strategies except for SCD and CHT pathways, which were sensitive to the changes in the length of non face-to-face contacts between the health care staff and the parents (Appendix A). Furthermore, parents indicated that the use of telecommunications offered an acceptable alternative for the delivery of positive NSB results. In terms of the methods used, parents viewed receiving the results via online meeting platforms such as Zoom and/or Microsoft Teams positively as it offered the opportunity for an immediate virtual face-to-face meeting with members of the clinical team and was, therefore, viewed as more personal than a telephone call. Additionally, this approach enabled the family to be contacted by condition specific specialist(s), which was viewed particularly favourably. For example, parents felt reassured that any questions they had about their child’s result could be answered immediately and by an expert. Indeed, the person delivering the positive NBS result and their condition specific knowledge were generally viewed as more important than whether or not the result was delivered in person or via telecommunication methods.

Our study has several limitations, of which the most important are the exclusion of (i) health outcomes, (ii) the cost and consequences of delivering false positive results and (iii) the training cost needed for implementing the new communication pathways. While considering the elements (i–iii) would be crucial for a full-economic evaluation, the objective of our research was to value the resources currently used in the existing communication pathway and those possibly needed if the new communication pathways were implemented. Future economic evaluations relying on these estimates should consider these limitations and test the impact of elements (i–iii) on their study conclusions. Nonetheless, our study is the first to address the variability of the communication pathways by proposing standardised, innovative, co-designed communication strategies to deliver the NBS results to parents. Another strength of our analysis lies in the data used, which was collected from all the centres processing the NBS and, thus, is a representative estimate of the costs of delivering NBS results in England. 

## 5. Conclusions

Poor, or inappropriate, communication strategies for positive NBS results can have a detrimental impact on parental outcomes in the short-term and can have a long-term impact on children and families [3,4,5,6,7,20,21]. In particular, the long-term impact of suboptimal delivery of NBS results may lead the parents to reduce their child’s interaction with others and worsen their relationship with the child. While these detrimental outcomes will impact children’s and parents’ health, this will come at some cost, which will mainly fall on the NHS budget. Likewise, these outcomes might result in parent productivity loss and, thus, future research should consider not only the cost for the NHS but also the possible costs falling on society, such as productivity loss.

While our study showed that implementing an innovative communication strategy would require a limited amount of healthcare resources, future research will need to prove whether this investment represents a good use of the NHS budget by showing its cost-effectiveness and implications at the society level.

## Figures and Tables

**Figure 1 IJNS-08-00019-f001:**
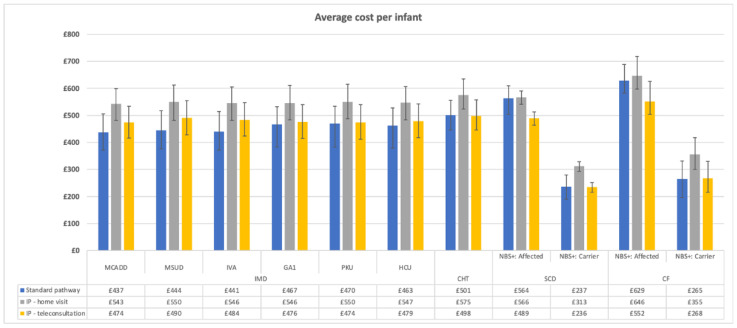
The average cost per infant by disease and communication pathway.

**Figure 2 IJNS-08-00019-f002:**
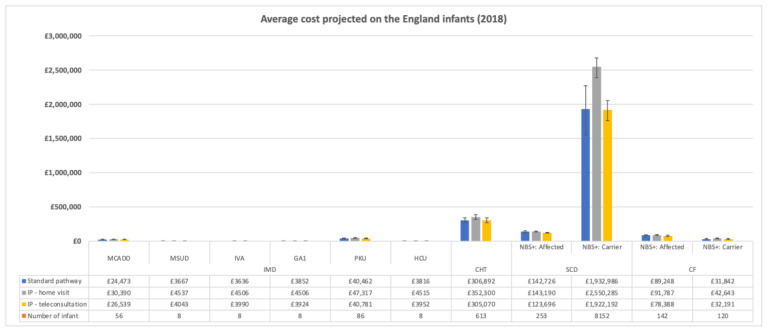
Projection of the average cost per infant by disease and communication pathway on the UK newborn population in 2018.

**Table 1 IJNS-08-00019-t001:** Cost of health care staff.

	Time (Minutes)	Cost per Minute	Cost per Contact	Source of Cost per Minute
Clinical nurse specialist				
home-visit ^+^	88.33	£1.18	£103.87	PSSRU 2019 [17]
surgery/hospital visit	55	£1.18	£64.68	PSSRU 2019 [17]
not face-to-face external interaction	22.81	£1.18	£26.83	PSSRU 2019 [17]
not face-to-face internal interaction	15	£1.18	£17.64	PSSRU 2019 [17]
Health visitor/midwife				
home-visit ^+^	88.33	£1.18	£103.87	PSSRU 2019 [17]
surgery/hospital visit	55	£1.18	£64.68	PSSRU 2019 [17]
not face-to-face external interaction	22.81	£1.18	£26.83	PSSRU 2019 [17]
not face-to-face internal interaction	15	£1.18	£17.64	PSSRU 2019 [17]
Consumables				
Leaflet		£2.50	£2.50	Assumption
GP				
home-visit ^+^	88.33	£3.65	£322.82	PSSRU 2019 [17]
surgery/hospital visit	55	£3.65	£201.00	PSSRU 2019 [17]
not face-to-face external interaction	22.81	£3.83	£87.37	PSSRU 2019 [17]
not face-to-face internal interaction	15	£2.30	£34.49	PSSRU 2019 [17]
Consultant (hospital-based)				PSSRU 2019 [17]
home-visit ^+^	88.33	£1.64	£144.77	PSSRU 2019 [17]
surgery/hospital visit	55	£1.64	£90.14	PSSRU 2019 [17]
not face-to-face external interaction	22.81	£1.64	£37.39	PSSRU 2019 [17]
not face-to-face internal interaction	15	£1.64	£24.58	PSSRU 2019 [17]

^+^ Home-visit includes travelling time to reach family residences.

## Data Availability

The data used in this publication is available in the Appendix A.

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
