# Peer review of "Delivering Positive Newborn Screening Results: Cost Analysis of Existing Practice versus Innovative, Co-Designed Strategies from the ReSPoND Study"

_2409-515X, 2022, doi:10.3390/ijns8010019_

Round 1
Reviewer 1 Report
The manuscript by Fusco, et al., presents a novel way to examine costs associated with the provision of positive newborn screening results. However, there are areas that could use improvement to better convey the outcomes and messages. Please see below for suggestions:
1) Introduction: While I understand the focus of this manuscript to be on the economic viability of implementing new interventions, there is no discussion on how these interventions actually improve the patient experience and/or save time and money downstream (e.g., does better communication with notification save downstream time and effort on parents and healthcare providers later in the diagnostic process)? Do they improve the patient experience in terms of understanding the results and next steps?
2) Materials and Methods: It states that the interventions were co-designed by parents who had received a positive NBS result - were these of cases that were ultimately determined to be true positives or carriers only? Were parents of false positive results included? If not, why were false positive results excluded (especially when they represent the majority of cases?) Are there any limitations in not including them?
3) Materials and Methods: It states that there were 3 study sites in the text, but the Figures refer to 13 sites? Please clarify. Also, it would be helpful to understand of the 257 positive results at the 3 sites, how many of each of the 9 conditions were represented.
4) Figures: Please define acronyms (e.g., CNS, GP, HV, CHR, etc.) I am also not certain that Figures 1 and 2 are necessary for this manuscript. While they are interesting in that they show the wide range of variation in how notification is accomplished, I think they may confuse the reader and take away from the ultimate goal of the paper. These could be added as an Appendix.
5) Table 1. Please indicate what the + means next to home-visit
6) Discussion. Please clarify the statement "On average, using the new co-designed strategies would lead to a small increment of the cost per infant approximately £37 nationally per annum." Is this saying that implementation of the new strategies would increase cost by £37 per infant? What is that total? It seems that the abstract contains data that is not in the paper (e.g., total costs) and that the costs presented may not align. It is hard to follow the cost estimates and savings and this may be better presented in an additional table.
Author Response
Reviewer 1
The manuscript by Fusco, et al., presents a novel way to examine costs associated with the provision of positive newborn screening results. However, there are areas that could use improvement to better convey the outcomes and messages. Please see below for suggestions:
1) Introduction: While I understand the focus of this manuscript to be on the economic viability of implementing new interventions, there is no discussion on how these interventions actually improve the patient experience and/or save time and money downstream (e.g., does better communication with notification save downstream time and effort on parents and healthcare providers later in the diagnostic process)? Do they improve the patient experience in terms of understanding the results and next steps?
We appreciate the reviewer comment and their understanding of the manuscript focus. However, we also acknowledge the point made by the reviewer on the importance of highlighting the patient experience. Therefore, an additional paragraph has been added to the introduction that refers to the findings of the published process evaluation that was undertaken and the associated views of health care professionals and parents in relation to the co-designed interventions (lines 51-58 and 65-73).
The manuscript now reads:
“Poor, or inappropriate, communication approaches can have an impact on children and families in the short [3-6] and long term [7]. Studies suggest distress caused can manifest as arguments between couples and apportioning of blame [3,5], alteration of life plans and inability to conduct tasks of daily living such as going to work or socialising [3], long-term alterations in parent-child relationships and mistrust and lack of confidence affecting ongoing relationships with staff. [7] There is also evidence of parents reducing their child’s interaction with others, particularly in the case of
CF.[3]
(omitted)
A process evaluation of the co-designed interventions, when used in practice in three National Health Service (NHS) provider organisations in England served by two NBS laboratories (NBSLs) that had been involved in the co-design process was undertaken.[11] Health care professionals who had used the co-designed interventions provided positive feedback in relation to their purpose, ease of completion and felt they were useful in terms of enhancing communication of positive newbornbloodspot screening results. Parents expressed that they felt the co-designed interventions improved consistency, pacing and tailoring of information and ensured reliable information was provided to families fol-lowing communication of the positive NBS result.”
2) Materials and Methods: It states that the interventions were co-designed by parents who had received a positive NBS result - were these of cases that were ultimately determined to be true positives or carriers only? Were parents of false positive results included? If not, why were false positive results excluded (especially when they represent the majority of cases?) Are there any limitations in not including them?
Thank you for alerting us to this. You are correct, this would have included the true and false positives since this refers to the communication of the initial positive NBS result i.e. before confirmatory testing has been undertaken. The text has been amended accordingly and now reads (lines 141-144):
“Then, we applied these costs to the number of infants with an initial positive NBS result for each of the nine conditions (this included true and false positives) to project the costs of implementing the communication pathways at the national level in England in 2018.[15]” We strongly agree that the monetary, and health related, consequences of delivering false positive results should be considered in the full economic evaluation. However, the current study is focussing on the cost of each communication pathway of NBS results. Thus, the presence, or absence, of false
positives is not influencing our results. While we will definitely consider this point in the next study (i.e. cost-effectiveness analysis), the current study does not apply to actual cases and, thus, we could not implement the suggested amendments. Nevertheless, we appreciate that these considerations are not straightforward and the readers may benefit from explicitly reporting this limitation of our study.
Thus, we updated the manuscript that now reads (lines: 290-297):
“Our study has several limitations, of which the most important are the exclusion of (i) health outcomes, (ii) the cost and consequences of delivering false positive results alongside (iii) the training cost needed for implementing the new communication pathways. While considering the elements (i-iii) would be crucial for a full-economic evaluation, the objective of our research was to value the resources currently used in the existing communication pathway and those possibly needed if the new communication pathways were implemented. Future economic evaluations relying on these estimates should consider these limitations and test the impact of elements (i-iii) on their study conclusions.”
3) Materials and Methods: It states that there were 3 study sites in the text, but the Figures refer to 13 sites? Please clarify. Also, it would be helpful to understand of the 257 positive results at the 3 sites, how many of each of the 9 conditions were represented.
Apologies and thank you for pointing this out. The way this was written was confusing and this has now been clarified. Baseline (existing) costs were calculated using data collected from all 13 study sites. Co-design of the interventions was undertaken in 3 of these study sites (served by 2 NBSLs).
The manuscript now reads (lines 96-102):
“There are thirteen NBSLs in England (study sites). Data were collected from all NBSLs regarding current communication strategies to provide baseline data for existing costs associated with communication of positive NBS results.[12] The interventions were co-designed by parents and health care staff in three National Health Service provider organisations (Trusts) in England served by two Newborn Screening Laboratories (NBSLs) that process comparable numbers of positive NBS reports annually for each of the nine conditions currently included in the NBS programme.[10]”
4) Figures: Please define acronyms (e.g., CNS, GP, HV, CHR, etc.) I am also not certain that Figures 1 and 2 are necessary for this manuscript. While they are interesting in that they show the wide range of variation in how notification is accomplished, I think they may confuse the reader and take away from the ultimate goal of the paper. These could be added as an Appendix.
We are grateful for the reviewer comment, and we acknowledge that their suggestions should be implemented. Thus, we specified the acronyms and moved the figures in appendices A and B.
5) Table 1. Please indicate what the + means next to home-visit
Apologies, this was an oversight and has now been amended by adding the necessary footnote in table 1.
6) Discussion. Please clarify the statement "On average, using the new co-designed strategies would lead to a small increment of the cost per infant approximately 37 nationally per annum." Is this saying that implementation of the new strategies would increase cost by 37 per infant? What is that total? It seems that the abstract contains data that is not in the paper (e.g., total costs) and that the costs presented may not align. It is hard to follow the cost estimates and savings and this may be better presented in an additional table.
After reviewing the manuscript, we agree with the reviewer that the calculation of the average cost per infant was not clearly reported. Therefore, we amended our manuscript, which now reads (lines 230-4):
“Assuming that the intervention pathways will be delivered to parents via home-visits and teleconsultation evenly (i.e. 50% of the results delivered via home-visits and the remainder via teleconsultation), the average cost per infant of the intervention pathways is 484, which is 37 larger than standard communication pathways average cost.”
Likewise, we also noticed that the text in the results section did not make prominent the data describing the total cost. We obviated this lack of clarity by adding a short sentence pointing the reader to these figures. Now the manuscript reads (lines 237-9):
“The averages of the costs at national level per each communication pathway are displayed in figure 2 alongside the number of infants per each disease.”

Reviewer 2 Report
This study aims to compute the mean costs associated with implementing the co-designed interventions during home-visits or via telecommunications compared to existing pathways per infant with a positive test result and project these costs to the national level across the English NHS per year. However I have the following concerns.
- The authors did not address the unit of the cost properly. It is difficult to understand if the authors are talking about "unit cost", "mean cost", "total cost" etc. when just the word "cost is used". Example of such "cost":
- Heading in Table 1: cost of health care staff (also, is it the mean time shown?)
- Line 203: cost at each study site
- Line 229: cost projected at the national level
- What is the base year of the currency?
- Appendices A to C show results from different centres. How is the cost per centre incorporated?
- Line 143: "positive result, namely those neonates carrying or affected by the relevant diseases". Does this mean that false positives are ignored? Why is that so?
- Line 203: "supplemental material 1-3". Please standardise the terminology - "appendix A to C" as stated in the supplementary material or "supplemental material 1-3" as stated in the main text.
- Appendix D: This wasn't mentioned in the main text. How should readers interpret this? In addition, is the unit "£"? There is also no appropriate heading for the table.
- What is the source for the number of neonates with each condition in 2018?
Can this information be presented as a table in the supplementary material please? - Why is the mean total cost to implement the different interventions for all 9 conditions in the NBS at the national level not studied?
- SCID is provided in some areas of England now as part of the NBS so why didn't the authors consider the scenario where SCID is implemented in all parts of England like the other 9 conditions?
Author Response
Please, see the attached file.

Reviewer 3 Report
This is an important study on the costs associated with different approaches to communicating newborn screening (NBS) results to parents in the UK. Given the expansion of NBS programs nationally and an increased emphasis parental communication and understanding this paper is both timely and extremely relevant. However, while this manuscript may be appropriate for publication in IJNS, there are a number of points that should be addressed by the authors before considering publication:
- While the paper focuses on describing the new co-designed communication approaches, it would be helpful for a bit more information on the current “standard of care” and its economic and potential social costs.
- The paper mentions the potential impact that communication of results may have on families but gives too little detail of what that impact or harm may be. Please elaborate a bit more on what those potential impacts or harms may be.
- While the co-designed approaches have been described in other papers, it would still be helpful to have a bit more detail on how these approaches are a departure from the NBS communication “standard of care”. The figures are helpful, but very complex, so some more description would be helpful.
- I am curious as to why the pathways indicate that primary contact is the mother, rather than parents more generally. What is done for adoptees or same-sex couples who have a surrogate. The language seems somewhat exclusionary.
- Overall, the discussion and conclusions section seem somewhat underdeveloped. It would be helpful to the reader to include more information on the potential policy and practice implications of these findings. For example, how might the implementation of the co-designed approaches practically impact the costs for NBS over time. While the overall costs may be incrementally lower or higher based on in person vs. telle-health, are there other costs for implementation that are not covered here. What were the limitations of this study?
- Because this study was conducted in 2017/2018 before the global pandemic, it would be helpful for the authors to contextualize their findings in a national and global health care environment, that looks very different that it did before COVID.
Author Response
Please, see the attached file.

Round 2
Reviewer 2 Report
I think the authors might have misunderstood some of my concerns.
- Regarding the base year of the currency, the authors should state that the cost is presented in which year instead of telling the authors which year the cost was obtained from because the year that the cost was obtained may not always be the cost presented for most studies as they could have inflated their cost to the latest year.
- “Delivering the NBS results using information communication technologies (i.e. teleconsultations) would lead to a small decrease in cost projected at the national level (i.e. the cost calculated by multiplying the costs per infant by the overall number of infants by disease) ... ” Are you referring to the "mean cost" or "total mean cost" etc at the national level? Likewise for the cost presented in the Abstract.
- Apologies, I had not realised that SCID screening started after the study was completed.
Author Response
I think the authors might have misunderstood some of my concerns.
1. Regarding the base year of the currency, the authors should state that the cost is presented in which year instead of telling the authors which year the cost was obtained from because the year that the cost was obtained may not always be the cost presented for most studies as they could have inflated their cost to the latest year.
We are grateful for the reviewer comment, which allowed us to improve the manuscript clarity.
Our current study presents the monetary values of resources used to deliver NBS results. To do so, we applied the most recent available unit costs (e.g. cost of 1 minute of work of a nurse) at the time of the analysis (i.e. 2019) to the resources used (e.g. 35 minutes of a nurse). As a result, we calculated the costs that would have been observed in 2019 (the most recent year available at the time of the analyses) – which does not need inflation.
While we believe this consideration could be not necessary for a health economists audience, we also acknowledge that this element might not be entirely clear to those readers having their expertise lying in other fields than health economics. Therefore, we updated the manuscript, which now reads (line:222-223)
“All monetary values obtained from our analyses represent costs that would have occurred between 2019 and 2020.”
2. “Delivering the NBS results using information communication technologies (i.e. teleconsultations) would lead to a small decrease in cost projected at the national level (i.e. the cost calculated by multiplying the costs per infant by the overall number of infants by disease) ...” Are you referring to the “mean cost” or “total mean cost” etc at the national level? Likewise for the cost presented in the Abstract.
We agree with the reviewer that the quoted passage could be improved by adding extra details on the metric considered in our analysis. Therefore, the manuscript now reads (line 314):
“(…) would lead to a small decrease in the total mean cost projected at the national level (…)”
3. Apologies, I had not realised that SCID screening started after the study was completed.
No comment is needed.
Reviewer 3 Report
I think the authors have done a nice job addressing the reviewer suggestions.
Author Response
I think the authors have done a nice job addressing the reviewer suggestions.
Thank you very much.